# Insights into the Role of Glucagon Receptor Signaling in Metabolic Regulation from Pharmacological Inhibition and Tissue-Specific Knockout Models

**DOI:** 10.3390/biomedicines10081907

**Published:** 2022-08-06

**Authors:** A Tate Lasher, Hemant Srivastava, Liou Y. Sun

**Affiliations:** Department of Biology, University of Alabama at Birmingham, Birmingham, AL 35233, USA

**Keywords:** glucagon receptor (GCGR), insulin sensitivity, glucose tolerance, metabolism

## Abstract

While glucagon has long been recognized as the primary counter hormone to insulin’s actions, it has recently gained recognition as a metabolic regulator with its effects extending beyond control of glycemia. Recently developed models of tissue-specific glucagon receptor knockouts have advanced our understanding of this hormone, providing novel insight into the role it plays within organs as well as its systemic effects. Studies where the pharmacological blockade of the glucagon receptor has been employed have proved similarly valuable in the study of organ-specific and systemic roles of glucagon signaling. Studies carried out employing these tools demonstrate that glucagon indeed plays a role in regulating glycemia, but also in amino acid and lipid metabolism, systemic endocrine, and paracrine function, and in the response to cardiovascular injury. Here, we briefly review recent progress in our understanding of glucagon’s role made through inhibition of glucagon receptor signaling utilizing glucagon receptor antagonists and tissue specific genetic knockout models.

## 1. Introduction

Glucagon is a 29 amino acid peptide secreted from the α-cells of the pancreatic islets of Langerhans first described by Kimball and Murlin in 1923 where pancreatic extracts produced hyperglycemic events upon administration on rabbits [1]. This 29 amino acid peptide is derived from the processing of proglucagon within the α-cells [2,3] by prohormone convertase 2 [4] and is subsequently secreted. From here, glucagon is able to exert its physiological effects on the body. While these effects have classically been limited to opposing insulin’s effects to maintain euglycemia, the role glucagon plays in amino acid and lipid metabolism as well as maintaining energy homeostasis has identified its effects as potential targets in the treatment of diabetes and obesity. 

Glucagon exerts its physiological effects through binding the glucagon receptor (GCGR) a G-protein coupled receptor (GPCR) successfully cloned by Jelinek and colleagues [5]. mRNA transcripts of the GCGR have been shown to be most abundantly expressed in the liver of rodents, with transcripts also being detected in the kidney, adrenal gland, heart, adipose tissue, spleen, ovary, pancreas, thymus, stomach, small intestine, thyroid, and skeletal muscle [6,7,8]. Additionally, conflicting reports exist on GCGR expression in the brain. Binding sites for glucagon have been reported in olfactory regions, hippocampus, anterior pituitary, amygdala, medulla, thalamus, and (to a lesser extent) hypothalamus when radiolabeled glucagon binding was investigated in rat brains [9]; however, the conclusion that these sites represent GCGRs is strained as more recent studies have demonstrated that glucagon is capable of binding the glucagon-like peptide-1 (GLP-1, another proglucagon-derived peptide secreted from L-cells located in the ileum and large intestine) receptor [10,11,12,13]. Svoboda and colleagues [7] reported detection of GCGR mRNA transcripts in the whole rat brain, but Hansen et al. [8] failed to detect GCGR transcripts in whole brain, cerebellum, cortex, or the pituitary of rats. More recently, the GCGR protein has been reported in the rodent hypothalamus and that its action may impair the ability of circulating glucagon to stimulate hepatic glucose production [14,15], indicating the existence of a physiologically relevant GCGR in the central nervous system. The varied distribution of GCGR expression would indicate a variety of functions and necessitates its study across specific tissues in order to completely understand glucagon’s physiological role. The development of GCGR-null and tissue-specific GCGR-knockout models in addition to the variety of pharmacological GCGR antagonists has advanced our understanding of glucagon’s effect throughout the body, and the purpose of this review is to summarize these findings as well as highlight gaps in the current body of research.

## 2. Systemic Effects of GCGR Signaling Ablation

Global deletion of the GCGR highlights glucagon’s role as a metabolic regulator beyond glycemic control. Mice lacking the GCGR display improved glucose tolerance, reduced plasma glucose levels in fed or fasted conditions, elevated pancreas mass, increased pancreatic glucagon content and proglucagon derived peptides, and dramatically increased blood glucagon content [16,17]. Additionally, GCGR-null mice have a marked increase in insulin sensitivity, requiring a significantly greater glucose infusion to maintain euglycemia in a hyperinsulemic clamp setting, with a reduction in hepatic glucose production necessitating additional glucose to overcome insulin’s actions being a likely cause for this observation. In the same study, glucose stimulated insulin secretion in islets isolated from GCGR-null mice was reduced in high glucose media as well as in response to non-glucose secretagogues. Additionally, in-vivo insulin secretion in response to glucose was dramatically elevated, with GLP-1R blockade only partially reducing this [18]. These studies establish that, while glucagon signaling indeed plays a major role in regulating glycemia, the activity of the GCGR also plays a role in regulating the production and secretion of glucagon and insulin. Christine Longuet and colleagues [19] also demonstrated that mice lacking the GCGR display elevated plasma triglyceride and free fatty acid levels following a prolonged fast resulting from interrupted p38 MAPK and PPARα signaling and subsequent impaired fatty acid oxidation, leading to rapid development of a fatty liver despite a leaner body following high-fat feeding. GCGR knockout mice have also been shown to have elevated plasma levels of nearly every amino acid and pancreatic α-cell hyperplasia [20,21]. 

When pharmacological inhibition is employed to disrupt GCGR signaling, many of the same physiological alterations are observed. Reductions in postprandial and ad-libitum glycemia, elevations in plasma glucagon, GLP-1, GLP-2, amino acid, and reduced plasma urea levels have been reported in a variety of lean, obese, and diabetic rodent models [22,23,24,25,26,27,28,29,30,31,32,33,34,35,36,37,38,39,40]. The resulting improvements in glucose homeostasis and insulin sensitivity resulting from these effects have identified the GCGR as a target for the treatment of obesity, type 1, and type 2 diabetes. While GCGR antagonism consistently improves glucose tolerance and insulin sensitivity [22,23,25,26,27,30,34,41,42] and stimulates weight loss through elevation of energy expenditure and reduced food intake [30,31,33,34], pharmacological inhibition of glucagon signaling has also been shown to dysregulate amino acid metabolism as reduced plasma urea and impaired amino acid clearance from the blood have been reported [35,37,38,43]. It is also noteworthy that pharmacological GCGR blockade resulted in the detection of glucagon in the cerebrospinal fluid of rats [34]. While the functional consequences of this are unclear, GCGR expression in the brain [14,15] would indicate that the GCGR blockade may have centrally mediated effects. It has recently been reported that central nervous system GLP-1 signaling can influence whole body glucose homeostasis and energy expenditure via a mechanism involving cross talk between the hypothalamus and pancreas [44], demonstrating that proglucagon-derived peptides can influence metabolism both centrally and peripherally. Additionally, reductions in respiratory quotient, representing metabolic shifts from glucose to toward lipid oxidation, have been reported in mice treated with GCGR antagonists [23,33,41] as well as reduction in plasma non-esterified fatty acids and reduction in various ceramide species being reported in diabetic mice administered a GCGR antagonist [34], demonstrating that blocking GCGR action has notable metabolic consequences outside of glucose homeostasis. 

Taken together, these provide evidence that, in addition to glucose homeostasis, GCGR action plays a considerable role in lipid metabolism, amino acid metabolism, and pancreas development. Importantly, many of the same metabolic changes observed in mice with genetic GCGR ablation are also observed in models where pharmacological GCGR antagonism is employed, demonstrating that these changes occur rapidly in the absence of GCGR action.

## 3. Hepatic Effects of GCGR Inhibition

Hepatic GCGR signaling has been shown to strongly influence control of blood glucose homeostasis. Mice with a Cre/LoxP mediated deletion of the GCGR in the liver display decreased random blood glucose levels, improved glucose tolerance in response to glucose challenges (both oral and intraperitoneal), highly elevated plasma glucagon levels, and enhanced insulin sensitivity as evaluated by intraperitoneal insulin challenge or a hyperinsulemic-euglycemic clamp [45,46], and isotope tracing during the clamp setting has shown that pharmacological antagonism of the GCGR increases hepatic glucose oxidation, glycolysis, and TCA cycle anaplerosis in obese mice [23]. Additionally, pharmacological blockade of the GCGR has been shown to decrease mRNA and protein expression of the gluconeogenic enzymes glucose-6-phosphatase and phosphoenolpyruvate carboxykinase 1 in the liver [23,28,32,34] as well as reduced activation of the key gluconeogenic transcription factor CREB [34]. Interestingly, when Rossi and colleagues employed a chemogenetic model of hepatic Gi-GPCR (a family of GPCR responsible for inhibiting the activity of Gs-GPCRs such as the GCGR), some opposite effects were observed. Mice expressing a designer Gi-GPCR in the liver generated via adeno-associated virus displayed impaired glucose tolerance but enhanced glycolysis and gluconeogenesis when activated. The researchers proposed that this counterintuitive increase in hepatic glucose production despite GCGR inhibition was the result of JNK pathway activation stimulating hepatic glucose production [47]. This shows that, apart from being a master regulator of metabolism itself, the hepatic GCGR is also involved in crosstalk between different families of GPCRs and that certain signaling pathways can compensate for the absence of hepatic GCGR signaling. These reports confirm hepatic GCGR signaling as an important component for the maintenance of euglycemia and that the interruption of GCGR action heavily influences hepatic glucose output.

Aside from glucose homeostasis, mice lacking the hepatic GCGR also display important differences in lipid metabolism. Hepatic GCGR deletion results in reduced fat mass accumulation in response to high-fat diet feeding [48], and obese mice with knockdown of the hepatic GCGR display lower levels of total cholesterol and low-density lipoprotein (LDL) cholesterol in the plasma [49,50]. However, the mechanism for this change is unclear. Spolitu and colleagues reported that adeno-associated virus mediated hepatic GCGR knockdown reduced the LDL-receptor expression by regulating the stability of PCSK9, responsible for directing the LDL-receptor towards lysosomal degradation [49]. When pharmacological GCGR antagonism is employed, inconsistent effects on plasma cholesterol have been reported, with both elevations [26,28] and reductions [25] in total and HDL cholesterol being observed. It is also noteworthy that GCGR antagonism has been reported to decrease abundance of several species of ceramides [34], but any physiological effects of these changes have yet to be described. Han and colleagues did not detect any differences in LDL-receptor or PCSK9 expression in small-interfering RNA mediated hepatic GCGR knockdown [50] but reported that increases in lipogenesis genes were likely responsible for the increases in total and LDL cholesterol. Consistent with this, Guan and colleagues detected elevated cholesterol absorption following GCGR antagonism; however, no changes in liver PCSK9 or LDL-receptor expression were detected [28]. Further complicating the interpretation of these studies is the paradoxical impact of GCGR agonism on body composition. A recently developed GCGR agonist with decreased affinity for the GLP-1R compared to glucagon has been shown to reduce fatmass, with this effect being mitigated in Cre/LoxP mediated hepatic GCGR-KO mice and without significant reductions in food consumption [51]. Taken together, these studies show that the hepatic GCGR impacts lipid homeostasis but that mechanisms of its control require further elucidation. 

Additionally, there is evidence that the hepatic GCGR plays a role in regulating development of pancreatic α-cells. Mice with a Cre/LoxP mediated hepatic GCGR deletion show α-cell hyperplasia [45], similar to that of GCGR-null mice reported by others [17,20,21], indicating that the mechanism for this observation is likely hepatic in origin. A marked increase in the gene encoding the glutamine transporter SLC38A5 has been reported in isolated α-cells of lice lacking the GCGR [52]. Additionally, Dean and colleagues reported that zebrafish with CRISPR-mediated *slc38a5b* knockout display reduced α-cell hyperplasia when the GCGR is globally deleted, and led them to hypothesize that alterations in circulating glutamine levels resulting from GCGR signal interruption led to α-cell proliferation [52]. This liver- α-cell axis is likely multifaced, however, as Galsgaard and colleagues have reported that glutamine failed to elicit glucagon secretion from perfused pancreata, where cystine, alanine, arginine, and proline did [37], with GCGR antagonism reducing the clearance of these amino acids. Interestingly, GCGR antagonism increased the survival of mice lacking ornithine transcarbamylase, a necessary urea cycle enzyme [40], suggesting that these alterations in amino acid metabolism may provide a promising target for treatment of urea cycle related disorders. These studies show that hepatic GCGR signaling can affect extrahepatic tissues by alteration of circulating amino acids and that the role of GCGR signaling extends beyond glycemic control. 

## 4. Pancreas-Specific GCGR Knockouts

Apart from being a simple regulator of glycemia, glucagon has also been shown to play important roles in regulating insulin production/secretion from the pancreas. In a study where transgenic mice overexpressing the β-cell GCGR were generated using the rat insulin promoter strategy, transgenic mice displayed elevated insulin secretion in response to glucagon challenge, and islets from transgenic mice displayed elevated insulin secretion in response to glucose treatment or glucose/glucagon co-treatment, as well as increased β-cell volume [53]. In the same study, Gelling and colleagues report elevated serum glucagon and lower hemoglobin A1c in the transgenic mice following three months of high fat diet feeding. In mice with a tamoxifen-induced β-cell GCGR knockout, there was no difference in insulin secretion compared to wild-types upon pancreatic perfusion of glucose and glucagon [13]. The same study found that glucagon was able to activate the GLP-1R, consistent with previous reports, and that perfusion of wild-type pancreas with glucose and a GLP-1R antagonist resulted in reduced insulin secretion. When pancreata from mice were lacking, the β-cell GCGR were perfused with glucose and a GLP-1R antagonist, dramatically reducing insulin production. Similar decreases in insulin secretion were seen when the pancreata of mice with a tamoxifen-induced β-cell knockout of the GLP-1R were perfused with glucose and a GCGR antagonist [54]. Recently, β-cells treated with glucagon and a GLP-1R antagonist at physiological concentrations of glucose display approximately 1/3 the insulin release compared to β-cells with intact GCGR and GLP-1R signaling [55]. These studies show that intraislet communication is essential for insulin secretion and that glucagon’s action at the GLP-1R significantly impacts insulin release and that the β-cell GCGR plays a non-negligible role in insulin release as well. 

Zhu and colleagues employed a chemogenetic approach to study GCGR signaling in pancreatic α-cells. Mice with a designer Gi-GPCR specifically expressed in α-cells generated using a tamoxifen-inducible Cre recombinase strategy displayed nearly no glucagon secretion following glucose or insulin challenges after activation of this designer receptor. In the same study when pancreata from designer Gi receptor mice were perifused with glucose and amino acids after receptor activation, there was no insulin secretion and glucose tolerance in these mice was reduced; however, insulin secretion was restored when exogenous glucagon was added. The same study found that wild-type insulin release was inhibited only after pharmacological blockade of both the GLP-1R and the GCGR or the GLP-1R in perifused islets, but that blockade of only the GCGR did not affect insulin release [56]. The discrepancy between these findings and those of Zhang and colleagues [55] is likely due to different concentrations of glucose used to stimulate the islets. This suggests that intraislet GCGR signaling is vital for glycemic control and suggests the crosstalk between glucagon secreted from α-cells and the β-cell GCGR and GLP-1R are necessary for the physiological response to circulating amino acids as well as glucose, and that the role each receptor plays is dependent on circulating glucose concentration. A visual summary of this crosstalk is provided in Figure 1.

GCGR signaling in the pancreas plays a role in pancreatic development in addition to islet function. The changes in the circulating hormone profile observed in GCGR-null mice and mice with pharmacologically inhibited GCGR signaling have prompted investigation into developmental consequences of this ablation. The previously mentioned α-cell size observed in GCGR-null mice was unchanged between GCGR-null and wild-type littermates at one day of age; however, by six weeks of age, GCGR-null animals display increased pancreas mass attributed to α-cell hyperplasia [17]. Additionally, pharmacological antagonism of the GCGR is sufficient to induce α-cell hyperplasia even after short-term treatment [22,24,36,41,57]. This α-cell expansion is observed alongside elevated pancreas proglucagon gene expression, glucagon and GLP-1 content, and somatostatin-staining islet cells [26,27,32,36,39]. Interestingly, this GCGR-antagonist induced α-cell proliferation appears to be dependent on age and angiopoietin-like-4 (Angptl4) as the degree of α-cell proliferation following GCGR blockade in aged mice was notably lower than in young mice [58] and that mice with a deletion of the Angptl4 gene only display α-cell proliferation following GCGR inhibition after administration of exogenous Angptl4 [29]. Additionally, conflicting reports exist on the influence of GCGR blockade on β-cell proliferation. When the PANIC-ATTAC model of inducible β-cell apoptosis was employed, GCGR antagonism elevated β-cell number and area within pancreatic islets as well as pancreatic insulin content [59]; however, no changes in β-cell morphology were observed in young or aged mice [58], and several have reported reductions in β-cell size following GCGR antagonism in obese mice [24,36]. Taken together, these reports demonstrate that GCGR action can dramatically and rapidly influence the development of α-cells and the release of insulin from β-cells but that the impact GCGR signaling has on β-cell morphology requires further investigation. 

## 5. Adipose-Specific GCGR Signaling

Comparatively fewer studies examine GCGR signaling in adipose tissue using a tissue specific knockout model. Brown adipose tissue (BAT)-specific GCGR knockout mice generated using the Cre/LoxP system displayed no phenotypic differences compared to wild-type mice with respect to bodyweight, cold response, whole body energy expenditure, and glucose homeostasis despite glucagon administration increasing oxygen consumption in BAT ex vivo and BAT-specific GCGR KO mice displaying elevated plasma non-esterified fatty acids following a fast and refeed [60]. Global deletion of the GCGR has been shown to alter the histology of white adipose tissue (WAT) and BAT, decreasing both the size and number of lipid droplets in these depots [61]; however, in BAT-specific GCGR knockout mice, no difference in BAT histology was observed [60], suggesting that GCGR activity elsewhere may be responsible for these differences. Indeed, changes in circulating factors such as amino acids have emerged as likely candidates for the liver-α-cell interaction [45], and it follows that a similar alteration could be responsible for these changes. Pharmacological blockade of the GCGR has been shown to alter the gene expression profile of WAT in mice, causing elevations in genes associated with extracellular lipoprotein handling and VLDL cholesterol as well as a notable increase in Angptl4 gene expression and plasma content, and may contribute to the α-cell proliferation seen in cases of interrupted GCGR signaling [29]. When GCGR signaling was pharmacologically blocked, Angptl4 knockout mice did not display the characteristic α-cell expansion typically seen with GCGR antagonism [29], potentially indicating that circulating Angptl4 may be a candidate for the mediator of GCGR inhibition’s effect on adipose tissue function and morphology. Taking these findings together would suggest that, in adipose tissue, disruption of the GCGR would alter the metabolism through a mechanism involving non-adipose tissue, as adipose-specific GCGR knockout mice display little phenotypic difference from wild types. Further study is necessary to establish a mechanism for the precise role the GCGR plays in adipose tissue and the role that circulating proglucagon-derived peptides play in adipose GCGR signaling.

## 6. Cardiovascular GCGR Activity

As with adipose GCGR action, less effort has been focused on the elucidation of glucagon action in the heart and cardiovascular system, despite GCGR expression being identified in the heart [6,7,8]. Ali et al. reported that mice chronically treated with glucagon prior to an induced myocardial infarction (MI) dramatically reduced subsequent survival, an effect that was rescued when p38 MAPK activity was pharmacologically inhibited [62]. In the same study, a tamoxifen-inducible cardiomyocyte specific Cre/LoxP mediated GCGR deletion showed significantly higher survival following induction of an MI relative to wild-types, and glucagon treatment prior to induction of MI did not impact this increased survival. Metabolomic profiling in these hearts revealed reduced long-chain acylcarnitines in the hearts of cardiomyocyte GCGR-KO mice following induction of ischemia; however, a causal mechanism has not been fully described.

GCGR antagonism following myocardial infarction has been shown to improve cardiovascular function. Specifically, post MI mouse hearts displayed greater ejection fraction and fractional shortening indicating that GCGR blockade improves these markers of mortality following MI. Additionally, it was reported that these animals also display improvements in insulin sensitivity as insulin stimulated greater glucose oxidation in the post MI heart following GCGR blockade [63]. The same study also investigated several signaling pathways and found elevations in IRS and AKT phosphorylation in post-MI mouse hearts following GCGR antagonism as well as reduced p38 MAPK, p70S6K, and mTOR phosphorylation, which the researchers hypothesized to contribute to the enhanced insulin sensitivity and reduced cardiac hypertrophy observed in these animals. When GCGR antagonists were administered to diabetic mice, ventricle pressure and function was improved, rescuing the cardiovascular decline seen in cases of diabetes [34]. The same study investigated GCGR antagonism in mice with lipotoxic cardiovascular injury and reported similar improvements in ventricle function as well as elevated AMPK activation in these hearts, indicating improved lipid oxidation, as well as a reduction in ceramide profiles in the lipotoxic cardiovascular injury mice.

Taken together, these studies suggest that GCGR activity plays a role in regulating cardiovascular function, metabolism, and response to injury. Interestingly, GLP-1R receptor agonism has also been shown to regulate the response to oxidative damage in cells [64], demonstrating that several proglucagon derived peptides can regulate response to cellular injury. It remains to be investigated if the protection from cellular injury conferred by GCGR receptor blockade and GLP-1R agonism share a mechanism or act through different pathways. The changes in lipid profile in the hearts of mice treated with GCGR antagonists may also play a role in these observed changes; however, further study is needed to formulate a mechanism.

## 7. GCGR Activity in the Gut

Elevated plasma levels and pancreatic content of glucagon-like peptide-2 have been associated with larger villi in the gastrointestinal tract of mice lacking the GCGR gene [65], further demonstrating that organ development is impacted by GCGR signaling. Pharmacological inhibition of the GCGR has a noteworthy impact on the gastrointestinal tract. Diabetic mice treated with a GCGR antagonist display increased intestinal length and intestinal epithelial area compared to nontreated controls that are accompanied by increases in GLP-1 and glucose-dependent insulinotropic polypeptides (GIP, secreted from K-cells within the duodenum, jejunum, and ileum) producing cells as well as proliferating GLP-1 producing cells [42]. The same study found that, in a GLUTag L-cell line, BrdU staining revealed elevated cell proliferation, reduced, and increased times spent in the G0/G1 and S phases of the cell cycle, respectively, following GCGR antagonist incubation. Additionally, primary enterocytes isolated from mouse and humans treated with a GCGR antagonist display elevated proglucagon mRNA levels as well as increased GLP-1 content and secretion [42]. A later study by the same group revealed that GCGR antagonism elevated proglucagon mRNA and GLP-1 content in the jejunum, ileum, and colon of diabetic mice and increased GLP-2 content within the duodenum, jejunum, and ileum of the same mice [39]. The same increases in GLP-1 and GIP staining cells were seen, and GCGR antagonist treatment reduced caspase 3 GLP-1 double stained cells, indicating that GCGR blockade reduces apoptosis in the GLP-1 producing cells of the GI tract. When isotope tracing was employed during a hyperinsulemic-euglycemic clamp, GCGR antagonism resulted in reduced glucose uptake in the jejunum of obese mice [23]. These demonstrate that, beyond the commonly observed pancreatic α-cell expansion observed in the absence of GCGR signaling, expansion and subsequent metabolic alteration in the GI tract epithelium may also accompany these changes. As the ablation of GCGR signaling triggers in a circulating amino acid profile, which has been shown to impact pancreatic development, it remains to be seen if alterations in circulating factors are drivers for the reported changes in proglucagon-derived peptides within the gut as well. 

## 8. Conclusions and Future Direction

Recently developed models of tissue specific GCGR knockout and pharmacological antagonism have advanced the current understanding of glucagon physiology and helped expand its appreciation as a regulatory hormone, rather than a simple counter to insulin action. Here, we briefly reviewed GCGR activity in the liver, pancreas, adipose, cardiovascular system, and gastrointestinal tract highlighting the impact beyond glucose homeostasis (summarized in Table 1 and visually in Figure 2). Several important gaps remain in our understanding of glucagon signaling, necessitating further research into GCGR activity. Off-target effects of pharmacological GCGR antagonists represent a problematic confound for interpretation of studies where they are employed. Several commonly used GCGR antagonists often reported as “selective” have been reported to display antagonism of the GLP-1R as well in cell culture [66]. This may explain some of the inconsistencies between pharmacological and genetic models of GCGR signaling ablation, but the direct consequences of off-target effects remain to be assessed in vivo. 

Development of additional genetic GCGR null models provide promising new avenues for research. Kidney-specific GCGR knockouts could provide critical information about glucagon’s extrahepatic effects, as this is where the GCGR is most abundantly expressed outside the liver [6,7,8] and may also regulate the physiological response to the altered pool of circulating amino acids and urea. Additionally, these same studies report conflicting information with respect to the GCGR expression in various regions of the brain. Generation of brain region-specific GCGR knockout models would more completely elucidate glucagon’s role as a regulator of metabolism as this is a center for energy expenditure in mammals and serves to reconcile these inconsistencies. While binding sites for glucagon have been identified in the brain [9], it is unclear how many of these binding sites respond to glucagon exclusively compared to cross reaction between glucagon at the GLP-1R, or if there is a physiological role for this cross reactivity. Indeed, the cross reaction of glucagon at the GLP-1R has proven to be vital for pancreas function and dual (or triple) agonists of the GCGR and GLP-1R (and GIP receptor) are currently being studied for the treatment of metabolic disorders. Further investigation into the cardioprotective effects of GCGR signal blockade will also provide valuable insight for treatment of cardiovascular injury, as elucidation of glucagon’s mechanism of action here may reveal attractive targets for intervention. In addition, we will investigate the GCGR antagonism’s mechanism for expanding proglucagon expressing cells in the GI tract and expanding our understanding of how proglucagon-derived peptides influence extrahepatic development. It will be important to further develop novel tools for GCGR signal ablation as they help explain the complex role of glucagon within the hormonal milieu.

## Figures and Tables

**Figure 1 biomedicines-10-01907-f001:**
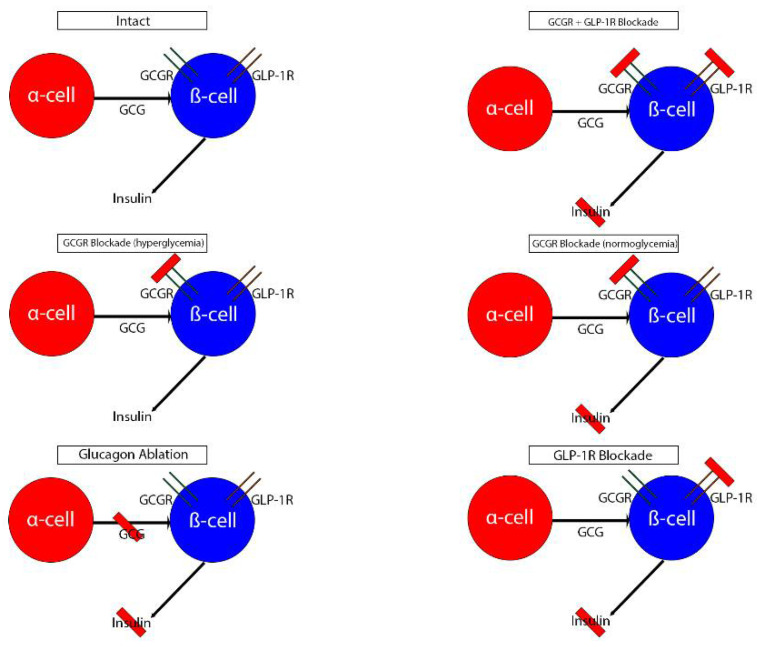
Visual summary of intraislet crosstalk between glucagon and insulin signaling. Abbreviations: GCG glucagon; GCGR glucagon receptor; GLP-1R glucagon-like peptide-1 receptor.

**Figure 2 biomedicines-10-01907-f002:**
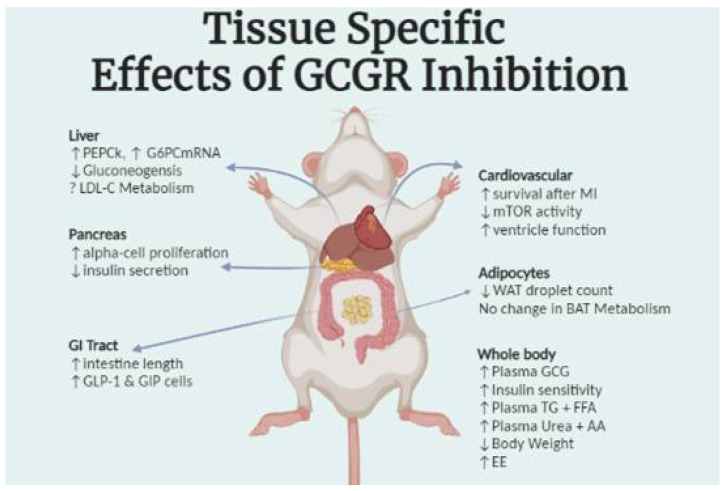
Visual summary of the physiological changes reported in various GCGR-KO models. Abbreviations: PEPCK phosphoenol pyruvate carboxykinase 1; G6PC glucose-6-phosphatase; LDL-C low density lipoprotein cholesterol; MI myocardial infarction; mTOR mechanistic target of rapamycin; WAT white adipose tissue; BAT brown adipose tissue; TG triglyceride; FFA free fatty acid; AA amino acid; EE energy expenditure.

**Table 1 biomedicines-10-01907-t001:** Overview of metabolic consequences resulting from GCGR-KO at specific tissues.

Tissue	Model	Insulin Signaling	Glucose Tolerance	Amino Acid Metabolism	Lipid Metabolism
Whole Body	Global GCGR-KO	↑ insulin sensitivity ↑ GSIS	↑ Glucose clearance	↑ Plasma AA	↓ Fatty Acid Oxidation
	GCGR antagonism	↑ Insulin sensitivity	↑ Glucose clearance	↑ Plasma AA	↓ NEFA ↓ Ceramides ↑ Fatty Acid Oxidation
Liver	Cre/LoxP Mediated KO	↑ Insulin sensitivity	↑ Glucose clearance	-	↓ LDL-C
	AAV-G_i_-GPCR	-	↓ Glucose Clearance	-	
	AAV GCGR Knockdown	-	-	-	↓ LDL-C ↓ Total-C
Pancreas	β-cell GCGR Overexpression	↑GSIS	-	-	-
	Tamoxifen β-cell GCGR KO	No change ↓ secretion (w/ GLP-1R antagonist)	No change	-	-
	Tamoxifen α-cell Gi-GPCR	↓ GSIS	↓ Glucose clearance	-	-
Adipose	Cre/LoxP Mediated BAT KO	No change	No change	-	↑ NEFA
Heart	Cre/LoxP Mediated KO	-	-	-	↑ long-chain Acylcarnitines

**Note:** GSIS—Glucose-stimulated insulin secretion; AA—Amino acid; NEFA—Non-esterified fatty acid; LDL-C—Low-density lipoprotein cholesterol; Total-C—Total cholesterol; GLP-1R—Glucagon-like peptide 1 receptor.

## Data Availability

Not applicable.

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
