# Peer review of "Insights into the Role of Glucagon Receptor Signaling in Metabolic Regulation from Pharmacological Inhibition and Tissue-Specific Knockout Models"

_biomedicines, 2022, doi:10.3390/biomedicines10081907_

Round 1

Reviewer 1 Report

Dear Authors,

Your work is of very good quality, it is well designed, composed and written. I found only few minor issues:

1. You use many abbreviations in the text. Some of them are explained directly in the text, some are well known (hopefully for all readers), but some require explanation, e.g. OTC (line 168). In my opinion it would be reasonable to add a list of abbreviations at the end (or at the beginning) of the paper.

2. Line 221: Angiopoietin instead of angiopotien-like-4

3. Line 301: gastric inhibitory peptide (GIP)... It is not the right abbreviation development. When the first incretin hormone was discovered in the late 1960s / early 1970s, that was how it was given its name. However, in later 1970s it was discovered that this hormone was responsible for stimulation of insulin secretion in a glucose-dependent manner, so the explanation of the acronym GIP has been changed to: glucose-dependent insulinotropic polypeptide. So, it has to be changed in the text. Moreover, it is reasonable to explain that GIP is secreted by K-cells located in duodenum, jejunum and proximal part of ileum, while GLP-1 secreting L cells are primarily found in the ileum and large intestine (colon).

4. You could also mention in the discussion that currently several GCGR antagonists and agonist (usually as dual agonists of GLP-1R and GCGR or triple agonists of GIPR, GLP-1R and GCGR) are studied for its usefulness in the treatment of obesity and diabetes.

Apart of these remarks I think that your work is valuable and surely worth to be published in Biomedicines after suggested corrections.

Best Regards

Author Response

Reviewer 1:
Dear Authors,
Your work is of very good quality, it is well designed, composed and written. I found only few minor issues:
1. You use many abbreviations in the text. Some of them are explained directly in the text, some are well known (hopefully for all readers), but some require explanation, e.g. OTC (line 168). In my opinion it would be reasonable to add a list of abbreviations at the end (or at the beginning) of the paper.
Response: We thank you for pointing out this source of confusion. We have included a list of abbreviations at the end of the manuscript.
2. Line 221: Angiopoietin instead of angiopotien-like-4
Response: We thank you for pointing out this error. It has been corrected in the manuscript.
3. Line 301: gastric inhibitory peptide (GIP)... It is not the right abbreviation development. When the first incretin hormone was discovered in the late 1960s / early 1970s, that was how it was given its name. However, in later 1970s it was discovered that this hormone was responsible for stimulation of insulin secretion in a glucose-dependent manner, so the explanation of the acronym GIP has been changed to: glucose-dependent insulinotropic polypeptide. So, it has to be changed in the text. Moreover, it is reasonable to explain that GIP is secreted by K-cells located in duodenum, jejunum and proximal part of ileum, while GLP-1 secreting L cells are primarily found in the ileum and large intestine (colon).
Response: We thank you for correcting our use of the antiquated terminology. The appropriate change has been made in the manuscript. We have also added the suggested explanations for GLP-1 and GIP secretion in the manuscript.
4. You could also mention in the discussion that currently several GCGR antagonists and agonist (usually as dual agonists of GLP-1R and GCGR or triple agonists of GIPR, GLP-1R and GCGR) are studied for its usefulness in the treatment of obesity and diabetes.
Response: We thank you for this constructive feedback of our discussion. Mention of dual and triple agonists has been included in the Conclusion and Future Direction section.
Apart of these remarks I think that your work is valuable and surely worth to be published in Biomedicines after suggested corrections.

Reviewer 2 Report

The authors have provided a brief review of glucagon receptor mediated effects on various aspects of metabolic function, as well as pancreatic development and cardiovascular function with an emphasis on genetic and pharmacological models of glucagon receptor inhibition. 

The manuscript has a number of shortcomings:

The grammar is less than ideal at times and there are multiple run on sentences

Many abbreviations are provided with no explanation (i.e., full name) and with little or no background concerning functionality of the indicated molecule, et cetera

Overall, the manuscript is not well-organized, and the arguments being presented are not particularly cohesive. This is especially true when the intrapancreatic crosstalk between glucagon and insulin signaling is being described.  A figure with an overview of the specific pathways indicated would be useful

There is no discussion of the potential confounds associated with many of the models that are described, especially pharmacological manipulation of glucagon signaling and potential off target effects

Given that the authors have no publication history in this area, it is unusual that they would have written a review over this topic. Some rationale or background concerning their own familiarity with the system is relevant and would be useful

Overall, the manuscript does little to add to our knowledge of glucagon biology and there are no future avenues of research that have been suggested to build on the ideas presented

Author Response

The authors have provided a brief review of glucagon receptor mediated effects on various aspects of metabolic function, as well as pancreatic development and cardiovascular function with an emphasis on genetic and pharmacological models of glucagon receptor inhibition. The manuscript has a number of shortcomings: Many abbreviations are provided with no explanation (i.e., full name) and with little or no background concerning functionality of the indicated molecule, et cetera Response: We thank you for your criticism. A table of abbreviations has been included at the end of the manuscript, and abbreviations have been clarified in the text. Overall, the manuscript is not well-organized, and the arguments being presented are not particularly cohesive. This is especially true when the intrapancreatic crosstalk between glucagon and insulin signaling is being described. A figure with an overview of the specific pathways indicated would be useful Response: We thank you for this criticism. A figure has been included, and the language has been clarified in this section. There is no discussion of the potential confounds associated with many of the models that are described, especially pharmacological manipulation of glucagon signaling and potential off target effects Response: We thank you for this constructive feedback. A discussion of off-target binding GCGR antagonists has been included.

Reviewer 3 Report

Comments to Authors              

            This study showed that: a) studies where pharmacological blockade of the glucagon receptor have been employed have proved similarly valuable in the study of organ-specific and systemic roles of glucagon signaling; b) studies carried out employing these tools demonstrate that glucagon indeed plays a role in regulating glycemia, but also in amino acid and lipid metabolism, systemic endocrine and paracrine function, and in the response to cardiovascular injury.

          Authors are kindly requested to emphasize the current concepts about these issues in the context of recent knowledge and the available literature. This articles should be quoted in the References list.

References

11.    Renoprotective Effect of Liraglutide Is Mediated via the Inhibition of TGF-Beta 1 in an LLC-PK1 Cell Model of Diabetic Nephropathy. Curr Issues Mol Biol. 2022; 44 (3): 1087-1114. Published 2022 Feb 25. doi:10.3390/cimb44030072.

22. Glucose-sensing glucagon-like peptide-1 receptor neurons in the dorsomedial hypothalamus regulate glucose metabolism. Sci Adv. 2022; 8 (23): eabn5345. doi:10.1126/sciadv.abn5345.

Author Response

This study showed that: a) studies where pharmacological blockade of the glucagon receptor have been employed have proved similarly valuable in the study of organ-specific and systemic roles of glucagon signaling; b) studies carried out employing these tools demonstrate that glucagon indeed plays a role in regulating glycemia, but also in amino acid and lipid metabolism, systemic endocrine and paracrine function, and in the response to cardiovascular injury.
Authors are kindly requested to emphasize the current concepts about these issues in the context of recent knowledge and the available literature. This articles should be quoted in the References list.
References
11. Renoprotective Effect of Liraglutide Is Mediated via the Inhibition of TGF-Beta 1 in an LLC-PK1 Cell Model of Diabetic Nephropathy. Curr Issues Mol Biol. 2022; 44 (3): 1087-1114. Published 2022 Feb 25. doi:10.3390/cimb44030072.
22. Glucose-sensing glucagon-like peptide-1 receptor neurons in the dorsomedial hypothalamus regulate glucose metabolism. Sci Adv. 2022; 8 (23): eabn5345. doi:10.1126/sciadv.abn5345.
Response: We thank you for your review of our manuscript. The works of Ninčević et al. (2022) and Huang et al. (2022) have been referenced in the text.

Round 2

Reviewer 1 Report

After corrections suggested by reviewers the manuscript can be accepted as is.

Best Regards

Reviewer 3 Report

Several improvements and clarifications have been made to the study.